# Lignin from Agro-Industrial Waste to an Efficient Magnetic Adsorbent for Hazardous Crystal Violet Removal

**DOI:** 10.3390/molecules27061831

**Published:** 2022-03-11

**Authors:** Rehab Ali, Zahwa Elsagan, Sara AbdElhafez

**Affiliations:** 1Fabrication Technology Research Department, Advanced Technology and New Materials Research Institute (ATNMRI), City of Scientific Research and Technological Applications (SRTA-City), Alexandria 21934, Egypt; 2Chemical Engineering Department, Faculty of Engineering, Alexandria University, Alexandria 21544, Egypt; zahwaelsagan90@gmail.com

**Keywords:** adsorption, crystal violet, magnetic adsorbent, kinetics, isotherm, thermodynamic

## Abstract

The presence of cationic dyes, even in a tiny amount, is harmful to aquatic life and pollutes the environment. Therefore, it is essential to remove these hazardous dyes to protect the life of marine creatures from these pollutants. In this research, crystal violet (CV) dye elimination was performed using a lignin copper ferrite (LCF) adsorbent. The adsorbent was synthesized and characterized using FTIR, Raman, SEM, EDX with mapping, and VSM, which proved the successful formation of magnetic LCF. Adsorption experiments were performed using different effective parameters. The highest adsorption potential (97%) was executed at mild operating conditions, with a 5 min contact time at room temperature and pH 8. The adsorption kinetic study utilized four kinetic models: first-order, second-order, intraparticle diffusion, and Elovich. The results revealed that the adsorption process complies with the pseudo-first-order with a maximum adsorption capacity of 34.129 mg/g, proving that the adsorption process mechanism is a physical adsorption process. Three isotherm models, Langmuir, Freundlich, and Temkin, were examined. The adsorption mechanism of CV onto LCF was also followed by the Langmuir and Freundlich models. The thermodynamic parameters were examined and revealed that the adsorption onto LCF was an exothermic process. It was proposed that the adsorption process is a spontaneous exothermic process. LCF appears to forcefully remove toxic CV dye from textile wastewater.

## 1. Introduction

Excessive acceleration in the world population, environmental contamination, fast diminishing of fossil fuels, effects of oligopoly in extraction and distribution of fossil fuels, and unstable fuel prices are the key factors leading to an urgent search for alternative, sustainable, and renewable energy resources [1,2]. Bioethanol is one of the most alternative renewable energy sources that has gained a lot of global attention in recent years, as it can be produced from agricultural waste. Annually, the world produces approximately 2.4 billion tons of agricultural waste, mostly disposed of incorrectly or open burned, leading to many environmental and health hazards [3]. The biomass mainly consists of lignin (15–30%), cellulose (30–50%), and hemicellulose (20–35%), which indicates that lignin is the second amplest component after cellulose. This polymer consists of complex phenyl propane units and is always produced as waste from bioethanol production and other industries such as the pulp and paper industry [4,5,6]. Hence, the annual lignin production can reach 100 million tons as agricultural waste and 50 million tons as industrial waste [7].

Water is used in various industrial and agricultural applications, and more than 359 billion cubic meters of wastewater are estimated to be generated per year worldwide due to toxic waste, dyes, petroleum, microorganisms, and/or chemicals [8]. All these substances hinder the advantageous use of water or the natural function of ecosystems and cause plenty of environmental and health obstacles [9]. Organic dyes are types of pollutants that are produced by the textile industry, as well as others, in significant amounts [10]. Annually, 10,000 different types of dyes formed are worldwide. The estimated percentage of dye discharged as effluent during the dyeing process ranges between 10 and 15% [8].

Consequently, more than 50,000 tons of organic dyes are released to the surroundings annually [8]. The presence of dyes such as congo red, red-blue 19, crystal violet, malachite green, or reactive red 141 in water, even in a small portion, is undesirable, as dyes hinder light penetration, destroy water quality, and ruin gas solubility. Moreover, they are highly toxic, endangering aquatic life [11].

Crystal violet (CV) is a triphenylmethane dye, abundantly used in fabric synthesis and biological coloring processes. However, it is severely carcinogenic and causes numerous health problems following human exposure, such as skin and eye irritation, kidney failure, respiratory system complication, nausea, vomiting, and permanent blindness. Therefore, CV removal from wastewater is crucial. However, CV is a compound with a convoluted structure that is difficult to remove. Several treatment methods have been investigated, including coprecipitation, adsorption, oxidation–reduction, flocculation–coagulation, several types of membrane filtration, and biological treatment [12]. Most of them require high investment and high operating costs, although they show low efficiency in dye removal. Currently, the adsorption process is a popular method that can be used as an ideal alternative to other expensive water treatment methods, attributable to its simplicity, selectivity, affordability, and extraordinary efficiency [13,14].

The high adsorbent synthesis cost, lengthy centrifugation, and/or filtration are the bottlenecks to using the adsorption process on a large or industrial scale. Therefore, many researchers have investigated the utilization of magnetic materials as adsorbents to simplify the adsorbent separation step, reduce operating time, and, consequently, operating costs. However, the high magnetic adsorbent synthesis cost can increase the overall adsorption cost. Hence, some studies have investigated the synthesis of magnetic adsorbents using different precursors such as carbon nanotubes [15], graphene oxide [16], and nanocellulose [17]; however, these precursors are nonrenewable, unfeasible for large-scale utilization, difficult to synthesize, and relatively expensive [18]. Thus, this work aims to utilize the lignin produced as waste from the bioethanol production of corn stover as a precursor to copper ferrite magnetic material to benefit from its renewability, availability, and low cost, and protect the environment from its improper disposal and pollution. This novel material, lignin copper ferrite (LCF), was utilized as a novel, effective, and affordable adsorbent for CV removal. The synthesized LCF was characterized using different techniques such as Fourier-transform infrared spectra (FTIR), Raman, scanning electron microscopy (SEM), energy-dispersive X-ray spectroscopy (EDX) with elemental mapping, surface charge (pH_zpc_), and vibrating-sample magnetometry (VSM). The selected adsorption parameters, such as operation contact time, LCF weight, initial CV concentration, temperature, pH, and ionic strength, were investigated to evaluate the adsorbent activity and optimize the adsorption operating conditions. Furthermore, the adsorption kinetics, isotherms, and thermodynamics were examined for adsorption of CV onto LCF. To the best of the authors’ knowledge, no study has previously investigated the induction of lignin copper ferrite or used it as an adsorbent.

## 2. Materials and Methodology

### 2.1. Material and Chemical Agents

Corn stover was obtained from El-Behara farms, Egypt. Crystal violet (C_25_H_30_Cl N_3_) was supplied from Bio-Basic Canada Inc., molecular weight 407.99 gm (assay 98%). Chemical materials such as sodium hydroxide (NaOH) with purity ≥ 97%, hydrochloric acid (HCl) (37%), ferric chloride (FeCl3), copper chloride (CuCl_2_), zinc chloride (ZnCl_2_), calcium chloride (CaCl_2_), sodium chloride (NaCl), and lithium chloride (LiCl) with purity ≥ 97% were provided from Sigma-Aldrich Co., St. Louis, MI, USA.

### 2.2. Adsorbent Synthesis

#### 2.2.1. Lignin Extraction

For lignin extraction, about 100 g corn stover was soaked in 1000 mL sulfuric acid (72%) for 4 h. This combination was treated in an autoclave at 120 °C for 1 h. Posteriorly, the combination was penetrated, and the formed solid was rinsed with warm water till pH 7. Then, the produced powder was located in a laboratory muffle furnace at 300 °C for 2 h and labeled TL300 for further use.

#### 2.2.2. Synthesis of Lignin Copper Ferrite

The iron and copper chloride salts were mixed in a molar ratio of 2:1 and added to 100 mL distilled water as a solvent in a three-neck flask. Then, the pH of this solution was adjusted at 11 using 2 N NaOH by titration under moderate stirring to ensure full contact and complete reaction. The suspension was left overnight at ambient temperature to separate and lie down. Then, the acquired mixture was filtrated and washed with distilled water three times. This prepared copper ferrite paste was mixed with 100 mL TL300 suspended solution with concentration 1% (*w*/*v*). The components suspended solution was placed in a 250 mL stainless steel Teflon lined autoclave and thermally treated at 120 °C for 12 h to significantly associate the reactants. Subsequently, the synthesized magnetic composite was allowed to cool. The mixture was filtrated, washed considerably with distilled water to eliminate residual ions, and sintered at 180 °C for 4 h in a vacuum oven. Finally, the created powder was calcined at 800 °C for 1 h and labeled LCF.

### 2.3. Characterization of LCF

Fourier-transform infrared (FTIR) detected the functional groups attached to the LCF surface (Shimadzu FTIR–8400 S, Kyoto, Japan). Raman spectra were recorded using (SENTERRA spectrometer, Bruker, Karlsruhe, Germany) with a 532 nm Ar laser. The LCF surface features and characteristic morphology were inspected by scanning electron microscopy using (SEM, JEOL Model JSM6360 LA, Tokyo, Japan) at room temperature with accelerating voltage 15 kV. The SEM device is equipped with an EDX detector to identify and map the synthesized LCF’s basic element structure. Magnetic characteristics of LCF were scrutinized by a vibrating-sample magnetometer (VSM, LakeShore-7410, Lake Shore Cryotronics, Inc., Westerville, OH, USA) with sensitivity up to 1 μ emu and a strong magnetic field up to ±20 ko_e_ to fully saturate the sample uniformly across the sample space.

### 2.4. Adsorption Investigates

A standard solution of the CV dye (100 ppm) was prepared and utilized for preparing the other investigated concentrations by dilution. Batch experiments were performed in a 250 mL Erlenmeyer flask placed on an orbital shaker at 200 rpm at 25 °C until the equilibrium was reached. A fixed amount of LCF was blended with 25 mL dye solution. After different time intervals, a specific quantity was suctioned from the reaction media by filter syringe, and residual CV dye concentrations were determined using a UV/visible spectrophotometer (7230 G, Shanghai, China) at maximum absorbance at wavelength (λmax = 570 nm), where this step was repeated three times. The effect of contact time (in a range from 1 min to 180 min), pH (2, 4, 6, 8, and 10), adsorbate solution concentration (5, 10, 25, 50, 75, and 100 ppm), adsorbent dose (ranged from 0.0125 to 0.1 g), and temperature (25, 40, 50, 60, and 70 °C) was inspected to determine the optimum conditions [9]. The empirical conditions of this process were listed in Table 1.

### 2.5. Zero-Point Charge (pH_PZC_)

The LCF surface charge was performed using the salt addition method by specifying the point of zero charges (pH_pzc_). Five solutions of 0.1 M NaCl with different pH values were adjusted between 2 and 12 using 0.1 M NaOH and 0.1 M HCl. A 0.25 g amount of LCF was mixed with 25 mL synthesis solution with shaking at 200 rpm for 24 h at ambient temperature. After this duration of shaking, the final pH values of the mixture were measured. Finally, the results of (pH_i_-pH_f_) were drawn against the pH_i_ to estimate the pH_PZC_ [11].

### 2.6. Ionic Strength

The influence in the presence of other salts on the removal efficiency of CV using LCF, known as ionic strength, was studied according to the following procedure. Three various concentrations (0.1, 0.5, and 1 M) of four various salts, ZnCl_2_, CaCl_2_, NaCl, and LiCl, were prepared by dissolving a certain amount of the salt in 25 mL CV (10 ppm). A 0.025 g amount of LCF was combined with the prepared solution and agitated for 1 h at 200 rpm. Subsequently, the adsorbent was filtrated, and a UV/visible device determined the dye concentration in the liquid phase.

### 2.7. Reusability Study

The reusability of the prepared magnetic adsorbent was determined through the adsorption–desorption of CV pollutants. Two desorbing agents, water and 0.1M HCl, were used to compare them. A 0.05 g amount of LCF magnetic adsorbent loaded with CV, after the adsorption process, was separated from the solution by a magnetic bar. The separated LCF was mixed with 25 mL eluent solution for 30 min at 200 rpm. After elution, the material was separated using the magnetic bar and reloaded with dye through the adsorption process. This process was repeated for 10 cycles, and the removal efficiency was computed after each cycle.

### 2.8. Removal Mechanism

Kinetic models are helpful in the examination of the mechanism and rate-limiting steps implicated in the adsorption treatment. The kinetic adsorption process for the CV onto the LCF surface was examined by fitting the experimental data to the pseudo-first-order Equation (1), pseudo-second-order Equation (2), intraparticle diffusion model Equation (3), and Elovich model Equation (4). The pseudo-first-order model is monitored by diffusion and mass transfer of the CV onto the LCF spot. The pseudo-second-order reveals that chemisorption is the rate-limiting step. The intraparticle diffusion model investigates the possible mechanism of adsorption of the studied molecules and verifies the transport mechanism. The Elovich model assumes that the existing solid surface is energetically heterogeneous.
(1)$$\ln(q_{e}-q_{t})=\ln q_{e}-k_{1}t$$
(2)$$\frac{t}{q_{t}}=\frac{1}{k_{2}q_{e}^{2}}+\frac{t}{q_{e}}$$
(3)$$q_{t}=k_{dif}\sqrt{t}+C$$
(4)$$q_{t}=\frac{1}{\beta_{e}}\ln(\alpha \beta )-\frac{1}{\beta_{e}}\ln t$$
where *q_t_* and *q_e_* are the mass of CV loaded on the unit mass of adsorbent at time *t* and equilibrium, respectively, (mg/g); *k*_1_ is the rate constant of the pseudo-first-order adsorption process (min^–1^); and *k*_2_ is the rate constant of the pseudo-second-order adsorption process (g/mg. min). *K_dif_* is the intraparticle diffusion constant (mg/g.min), *C* is a kinetic constant, and *α* is the initial rate (mg/g min) as (dq_t_/dt) approaches *β_e_* when *q_t_* is about zero. The parameter *β_e_* is the extent of activation energy and surface coverage for chemisorption (g/mg).

### 2.9. Isothermal Studies

Adsorption isotherms were studied for different initial CV concentrations (10–100 ppm) and room temperature, and adsorbent weight 0.025 g, which was added to 25 mL CV solution at constant shaking (200 rpm), was used to study the Freundlich, Langmuir, and Temkin isotherms. Three isotherm models, Langmuir Equation (5), Freundlich Equation (6), and Temkin Equation (7), were studied to describe the influence of the interaction between the CV and LCF to calculate the capacity of the LCF and detect whether the adsorption would be monolayer or multilayer [13,19].
(5)$$\frac{C_{e}}{q_{e}}=\frac{1}{q_{m}K_{L}}+\frac{C_{e}}{q_{m}}$$
(6)$$\ln q_{e}=\ln k_{f}+\frac{1}{n}\ln C_{e}$$
(7)$$q=\frac{RT}{b}\ln k_{T}+\frac{RT}{b}\ln C_{e}$$
where *C_e_* is the equilibrium CV concentration in the solution (mg/L), *q_m_* is the maximum adsorption capacity (mg/g), *k_L_* is the Langmuir constant that is related to the affinity of binding sites and corresponds to the energy of sorption (L/mg), *n* and *k_f_* are the Freundlich constants related to adsorption intensity and the adsorption capacity, *R* is the universal gas constant (8.314 J mol^–1^ K^–1^), *T* is the absolute temperature (K), and *K_T_* is the equilibrium-linked constant (mg^–1^).

### 2.10. Thermodynamics Studies

The thermodynamic behavior of the CV adsorption onto LCF was weighed by determining the change in Gibbs free energy (ΔG°), enthalpy (ΔH°), and entropy (ΔS°). These aspects were studied using the following Equations (8) and (9) [19].
(8)$$\ln\frac{q_{e}}{C_{e}}=\frac{\Delta S}{R}-\frac{\Delta H}{RT}$$
(9)ΔG=ΔH−TΔS
where ΔG°, ΔH°, and ΔS° are the deviations in free energy (KJ/mol), enthalpy (KJ/mol), and entropy (KJ/K.mol), respectively. T is the temperature of the adsorption system (K).

## 3. Results and Discussion

### 3.1. Adsorbent Characterization

The FTIR spectroscopy was performed to detect the functional groups that are attached to the LCF surface, as shown in Figure 1. There are three bands that emphasize the presence of lignin in the sample. The first band is at 2321 cm^–1^, which is related to the hydroxyl functional groups (–OH) in the stretching vibration mode [20,21]. The second band is at 2047.28 cm^–1^, which corresponds to the symmetric and asymmetric (C-H) stretching vibrations of the (CH_2_) and/or (CH_3_) groups related to lignin. The third band is at 1637 cm^–1^, which indicates the presence of the C=O stretching and C=C aromatic skeletal stretching vibrations of lignin [6]. The band at 1540 cm^–1^ corresponds to the C=C stretching of aromatic rings [22]. These bands are the main characteristics of lignin. The intensity of these bands is relatively small, as the lignin was mixed in a small amount of 1% (*w*/*v*). The band at 1080 cm^−1^ may be due to the presence of (Si-O-Si) [23]. The strong band at 528 cm^−1^ is related to the stretching vibration of the typical magnetic composite structure (Cu^+2^-O^–2^) octahedral group [24]. The FTIR results emphasize the successful formation of lignin copper ferrite. Moreover, the presence of hydroxyl groups on the LCF surface can improve the adsorption process by interacting and attaching the CV cationic dye onto the -OH groups on the LCF surface [21].

Raman spectra analysis was performed to provide further information for the synthesized composite. As shown in Figure 2, the Raman spectra were recorded between 50 and 4500 cm^−1^ at room temperature. The strong peak at 1312 cm^–1^ and the peaks at 606, 405, and 353 cm^–1^ are the main characteristics of lignin, as the elevation at 1312 cm^–1^ contributes to the aliphatic (O-H) bending of lignin [25]. The peak at 655 cm^–1^ corresponds to cubic-inverse cuprospinel A_1g_. The peaks at 576 and 493 cm^–1^ are assigned to cubic-inverse copper ferrite F_2g_ (1), F_2g_ (2). The peak at 289 cm^–1^ corresponds to E_g_ and/or E_1g_ of the inverse copper ferrite. The height at 223 cm^–1^ is related to the inverse cuprospinel F_2g_ (3) [8,26]. The Raman results are compatible with the FTIR results and prove composite components’ coalescence.

The presented SEM images of LCF in Figure 3 show that the synthesized LCF has a rough, irregular, and cracked surface, which may lead to a high contacting surface to develop the CV adsorption process and expose higher functional (-OH) group sites to the cationic dye.

The LCF elemental composition and mapping were detected using EDX analysis to verify the ratios between the three main components of the adsorbent. As shown in Figure 4a, the results show that the carbon content is about 0.8, related to the stoichiometric ratio to % lignin in adsorbent paste (1%). The mass percentage of iron is higher than the mass percentage of copper added stoichiometrically. Moreover, the EDX elemental mapping images of the LCF presented in Figure 4b–f represent the C, Fe, Si, Cu, and LCF samples, respectively. The elemental mapping distributed in the as-prepared LCF confirms the successful interaction of lignin in the synthesis of copper ferrite magnetic paste [24].

The magnetic hysteresis (MH) loops shown in Figure 5a were achieved at room temperature using the utilized field range ±20 kOe. The results demonstrated that the sample has magnetic properties due to the regular shape with a value of moderate saturation magnetization (Ms) = ±11.396 memu of the LCF. The packed surface morphology and high-iron ions percentage contributed to the magnetic properties’ presence [17]. As shown in Figure 5b,c, the adsorbent could be easily collected with the magnetic bar even if it was in an aqueous medium. Accordingly, the LCF could be easily collected and removed from the dye solutions after the adsorption process, showing easier operation due to the sensitive magnetic response [27].

### 3.2. Effect of Experimental Conditions

#### 3.2.1. Effect of Adsorption Contact Time and Kinetic Models

The effect of agitation time on both the % removal and amount of dye adsorbed was studied at different times varied from 0 to 15 min, dye concentration 50 ppm, and amount of adsorbent 0.05 g at room temperature with agitation rate 200 rpm. As shown in Figure 6, the amount of CV adsorbed on LCF versus contact time is initially quite rapid. Then, the adsorption efficiency starts to decrease. The highest adsorption efficiency reached about 68% after 5 min, which could be considered the optimum contact time. This rapid equilibrium points out the adsorption process pursues fast kinetics, where the CV expeditiously occupies the offered adsorption active sites on the LCF until they are saturated. This behavior may be attributed to unoccupied active sites at the adsorbent surface and the high difference between the CV concentration in solution and adsorbent surface. After 5 min, the free adsorption sites drop together with CV concentration. The gradual retardation of adsorption could be because of monolayer formation on the adsorbent surface due to the low availability of vacant sites after attaining equilibrium.

The fits of the kinetic records to the adsorption data are explored with pseudo-first, pseudo-second, intraparticle diffusion, and Elivoch models (Figure 7a–c). The kinetic parameters were calculated from the linearized integral for the four models and are summarized in Table 2. For CV, Lagergren’s model showcased considerable experimental values from the theoretical ones; generally, this line did not permit through the origin, suggesting that boundary-layer diffusion also exists. As listed in Table 1, the values determined using the pseudo-first-order model are similar to practical values than those expending from the other studied models. Its correlation coefficient (R^2^ = 0.9998) is closer to unity than the others, indicating that CV adsorption on LCF was a pseudo-first-order process. Hence, this process is supposed to be controlled by physical adsorption [9,28].

#### 3.2.2. Effect of Adsorbent Weight on CV Removal

The influence of adsorbent weight on CV removal was inspected by adding different adsorbent masses (0.0125, 0.025, 0.05, 0.075,0.1, and 0.125 g) in 25 mL 10 ppm dye solution, and contact time 5 min at room temperature with agitation rate 200 rpm was tested. As shown in Figure 8, the % removal of CV showed a gradual escalation from 36 to 75% as the adsorbent dosage was augmented from 0.0125 to 0.075 g. After that, an insignificant increase in adsorbent effectiveness occurred up to (78.7%). The increase in dye removal when the adsorbent weight was 0.075 g could be interpreted as increasing removal efficiency in the first stage caused by the small adsorbent amount for the efficient spreading of the nanoparticles in the dye solution; therefore, available extra surface area and more available active sites after the almost constant percentage removal has occurred in the following step may be due to the accumulation of the particle and blocking the active site, which obstructs the reached the CV and its contact to the active site. Therefore, the optimum dosage is found to be 3 g/L [29]. Similar trends have also been proposed by Cheruiyot et al. in their studies on the removal of crystal violet from aqueous solution using coffee husk as an affordable adsorbent [30].

#### 3.2.3. Effect of Initial Dye Concentration and Isotherm Models

The effect of the initial CV synthesis concentration on the adsorption process efficiency using the LCF played an important role in the adsorption process. As shown in Figure 9, the equilibrium adsorption capacity gradually increases from 4 to 27 mg/g, then this improvement almost linearly increases from 27 to 30 mg/g when the concentration of CV solution increases from 50 to 100 ppm. This phenomenon can be clarified by the interaction between CV ions and the adsorbent. Increasing the dye concentration will enhance the dynamic force by reducing the mass transfer resistance. In addition, the density of CV in the solution increases, and the ratio of CV ions to the accessible adsorption sites grows as well. Therefore, more CV ions in the solution can be adsorbed on LCF, giving the equilibrium adsorption capacity. Similar phenomena have been observed in the adsorption of crystal violet dye using water hyacinth plant biosorbent [31].

On the other hand, the % removal showed a different trend when it was compared with the adsorption capacity. The maximum % removal occurred with 5 ppm (80%), and by increasing the concentration, the % removal came down to reach 30%. However, the difference in trend between the q_e_ and % removal can explain that the rising initial CV concentration enhances mass transfer and the consequent absorption of the dye grains. Nevertheless, this increase is not proportional to the amount of dye, leading to a lessening in % removal.

Likewise, the analyzed experimental data of the studied three isotherm models exist in Figure 10a,b. Three isotherm models were tested to show the transfer mechanism of adsorbate species during the adsorption process between the liquid and solid phase in the equilibrium. The isotherms parameters are listed in Table 3. The sorption isotherms show the distribution of adsorbate molecules between the liquid and solid phase when the sorption experiment reaches an equilibrium state and determine the adsorbent capacity. Compared to Temkin, the R^2^ values of Langmuir and Freundlich were higher than (0.99). This R^2^ value infers that the Langmuir and Freundlich isotherms are compatible for explaining the adsorption mechanism of CV by LCF adsorbent. This isotherm could describe the adsorption process between the CV and LCF as multilayer adsorption with adsorption energy between adsorbent and dye molecules on the heterogeneous surface. Applying the Langmuir model, the monolayer maximum adsorption capacity for LCF was 32.25 mg/g and described the affinity of the binding sites K_L_ = 0.1411 L/mg.

#### 3.2.4. Effect of Temperature and Thermodynamics Outcome

The effect of temperature on the CV dye sorption on LCF was studied at various temperatures: 25, 40, 50, 60, and 70 °C. As shown in Figure 11, the adsorption capacity of CV decreases from 79 to 70% when the temperature increase from 25 to 70 °C, indicating an exothermic adsorption process. This deficiency could be due to the cracking of some of the formed bonds between adsorbent and adsorbate after raising the temperature. These results specify that it is preferable to operate this adsorption process at room temperature, which is economical and positively affects the adsorption process. This behavior is consistent with another study investigated by Al-Shehri et al. [32].

To evaluate the thermodynamic parameters, the ln (q_e_/C_e_) was plotted against 1/T (K^–1^) for adsorption at different temperatures that range from 25 to 70 °C. The initial CV concentration (10 ppm) was constant with 0.025 g adsorbent, 5 min contact time, and pH 7. The ΔH_0_ and ΔS_0_ were determined from the trend line slope and intercept (Figure 12). Conversely, ΔG_0_ was calculated from the values of ΔH_0_ and ΔS_0_ using Equations (8) and (9). The thermodynamic obtained parameters are listed in Table 4. It was found that ΔH_0_ was –1.3 KJ/mol for CV adsorption by LCF with a negative value, which agrees well with the outcome of Figure 11. Moreover, the ΔH equals –1.3 kJ/mol, smaller than 40 KJ/mol, implying that CV was physically adsorbed on the adsorbent active sites [28]. As listed in Table 4, the adsorption process is spontaneous at all studied temperatures, as the value of ΔG_0_ is negative. The positive sign of ΔS_0_ means that the system has a good affinity of CV towards magnetic LCF adsorbent [33].

#### 3.2.5. Effect of Adsorption pH and Point of Zero Charge

The pH of the solution is the most vital factor that controls the CV adsorption onto LCF particles. This importance is due to many reasons, e.g., the adsorption of H^+^, OH^−^, and other ions can affect the adsorption media pH. Moreover, the solution pH directly affects the CV degree of ionization, the dissociation degree of the functional groups on the LCF active sites, and the LCF surface charge [3]. Batch adsorption experiments were performed to investigate the influence of the solution pH on the CV adsorption onto the LCF in the solution pH range from 2 to 12. As shown in Figure 13, at acidic media, the solution pH leads to decreased CV removal as a result of the competition between the generated H+ and the CV cations ions [18]. However, the maximum CV removal (97%) occurs at solution pH 8. It is reported that the low pKa of the CV (0.8) and (5.3) leads to complete and facile ionization of this dye at almost all pH values and is present in the solution as cations [33,34]. Therefore, adjusting the pH solution to be higher than the pk_a_ value is favorable in dye removal. Increasing the pH from 2 to 8 increases the electrostatic interaction between the LCF negative charges and the CV dye ions [11]. Further, an increase in the solution pH up to 12 leads to a decrease in the CV removal to 85%, which may be owing to repulsion forces between OH- groups generated in the solution due to the high solution pH and the anionic function groups on the LCF surface. Cheruiyot et al. achieved optimum dye removal at pH = 8 with the same behavior of dye by using coffee husk [30].

As shown in Figure 14, the magnified LCF’s point of zero charges (pH_zpc_) was in the neutral range (6.1). The pH_pzc_ indicates integral neutralization; thus, at pH lower than 6.1, the surface of the LCF adsorbent is positively charged, and at pH above 6.1, it is negatively charged. Nair et al. informed a pH_pzc_ for a lignin–chitosan blend in nearly nutrient media (6.8) [22]. Since CV is cationic, adsorption experiments were preferable to perform at pH higher than 6.1, which was provided in the pH effect study.

#### 3.2.6. Ionic Strength

The consequence of the ionic strength on the CV removal using LCF was studied after dissolving four salts with the same negative radical (Cl^–1^) and different in the positive radical (Na^+1^, Li^+1^, Zn^+2^, and Ca^+2^). These cationic ions can alter the surface property. Moreover, the presence of ions competes with the hydrated ions and pollutants for the active sites on the material particles, and this competitiveness negatively affects waste removal.

As shown in Figure 15, after adding monovalent salts, the adsorption efficiency decreased from 97 to 71% and 82% due to adding 1 M (NaCl) and (LiCl), respectively. The hydrated radius of Na^+^ = 2.76 Å, which is shorter than Li^+^ = 3.4 Å. This clarifies the stronger competition between CV, cationic dye, and other positive radicals; as a result, the more pronounced impact on CV adsorption with the Li^+^ compared with Na^+^. However, the limited drop in the CV adsorption efficiency points out the selective adsorption of LCF for CV. As reported, the ions having a small hydrated radius forms a threat to the adsorption as the cation conquers more sites on the adsorbent [35]. A similar sequence has been reported for methylene blue adsorption onto multiparous palygorskite activated by ion beam bombardment [36]. In addition, comparing the adsorption capacity of 1 M salt solution with other lower concentrations to the adsorption capacity, it is found that minor results differences are yielded.

On the other hand, the divalent salts (ZnCl_2_ and CaCl_2_) with the hydrous radius of Zn^+2^ = 4.26 Å and Ca^+2^ = 4.1 Å followed the same trend as monovalent salts, but with fiercer competition. Figure 15 shows that the % removal decreased from 92 to 56.8 and 66% by dissolving 1 M of Zn and Ca salts in the dye solutions, respectively. The adsorption capacity increases for calcium and zinc salts by decreasing the salt concentration in the dye solution. Consequently, none of the previous salts achieved a higher removal percentage than the zero molar or even close. This behavior is due to the increase in the positive ionic charge, which accelerates the repulsion force with the cationic dye resulting in aggressive competition to occupy the active sites.

#### 3.2.7. Reusability of LCF

One of the most important economic aspects that is desirable to designate a material is the possibility of reusing it several times. Compared with several eluents, many studies have found that reusing acids as desorbing agents for removing cationic dye is the most effective. As shown in Figure 16, using HCl has a great positive effect on the resorption process, as LCF is able to reuse the five cycles with removal efficiency 86%, and this reaches 60% after 10 cycles. This trend could be attributed to the fact that the process of reabsorption takes place effectively where the hydrogen ions are contained, competing with the dye molecules and extracting them from the adsorbent surface and replacing it, which leads to the possibility of reusing the material many times. On the other hand, the use of slightly heated water resulted in less effectiveness than that previously studied, but it is resendable as well, as when the adsorption reached 80% after the 4th cycle, then this percentage declined to 47% at the 10th cycle. Perhaps the efficiency of the reuse process is due to the weak physical bond (van der Waals bond) formed between the adsorbent and adsorbate, which is easy to break. From the above, the LCF can be considered an economically effective magnetic adsorbent and can be applied to the industrial field, and it is recommended to be used in continuous systems with residual time 5 min to maximize the utilization of its properties. This indicates an important feature that allows it to be used in the industrial field effectively.

#### 3.2.8. Comparative Study of CV Dye Uptake Capacity with Different Adsorbents

Recently, using affordable adsorbents in wastewater treatment has been broadly investigated. The capacity of removal of CV and other dyes by LCF and other adsorbents is listed in Table 5. It turned out that as a result of the presence of negative functional groups in the LCF surface to remove CV, this adsorbent is superior to its peers in its ability to get rid of a high proportion (97%) of the pollutants exposed to it and this is a very little time that was not reached before, which endorses the novelty and importance of the prepared adsorbent in the current study among the other listed adsorbents.

## 4. Conclusions

In this research, lignin copper ferrite, a novel adsorbent, was successfully synthesized and characterized. It proved to be an efficient method to remove crystal violet dye from its synthesis solution. The impact of contact time, weight of adsorbent, initial dye concentration, temperature, and pH on this adsorption process efficiency was studied. It was found that the best process % removal (97%) in equilibrium time was 5 min, LCF amount 0.025 g, initial concentration 10 ppm, room temperature, and pH 8. Furthermore, it was found that this adsorption was deleteriously affected by a pH decrease from 8 to 2. Thermodynamic results showed the physical nature of CV by LCF adsorbent. In addition, the isotherm study showed that CV adsorption followed the Langmuir and Freundlich isotherm model. Furthermore, LCF could easily be regenerated and recycled, which makes it applicable for continuous treatment processes. The high maximum adsorption capacity of LCF (34.12 mg/g) will meet special interest in polluted water treatment, among other low-cost adsorbents. Further studies about the optimization of continuous adsorption conditions are running in our lab.

## Figures and Tables

**Figure 1 molecules-27-01831-f001:**
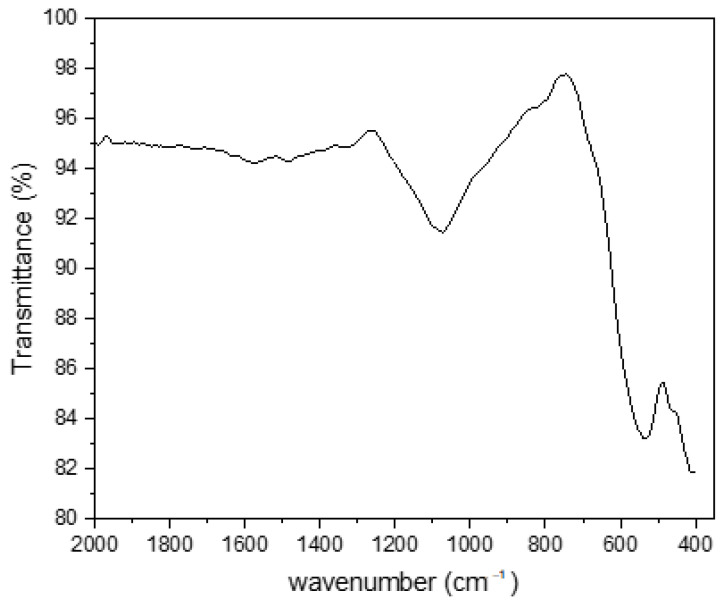
Fourier-transform infrared (FTIR) spectra of LCF.

**Figure 2 molecules-27-01831-f002:**
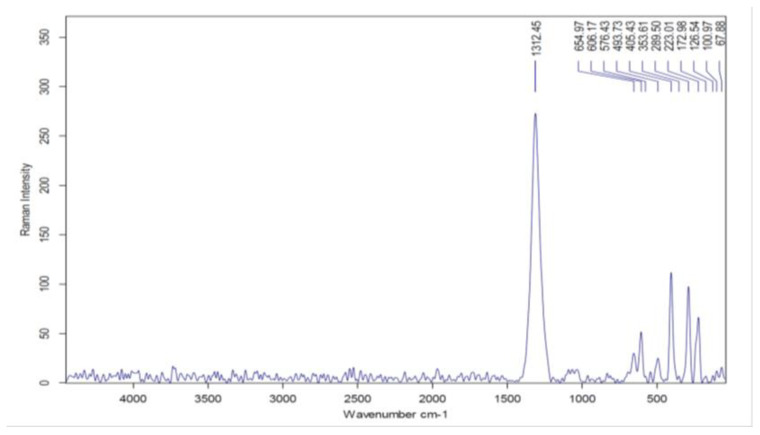
Raman spectra of LCF adsorbent.

**Figure 3 molecules-27-01831-f003:**
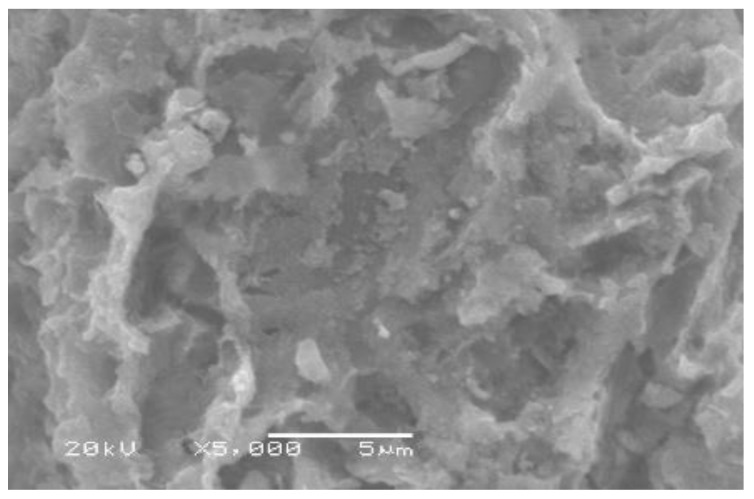
Image microscope photograph of LCF.

**Figure 4 molecules-27-01831-f004:**
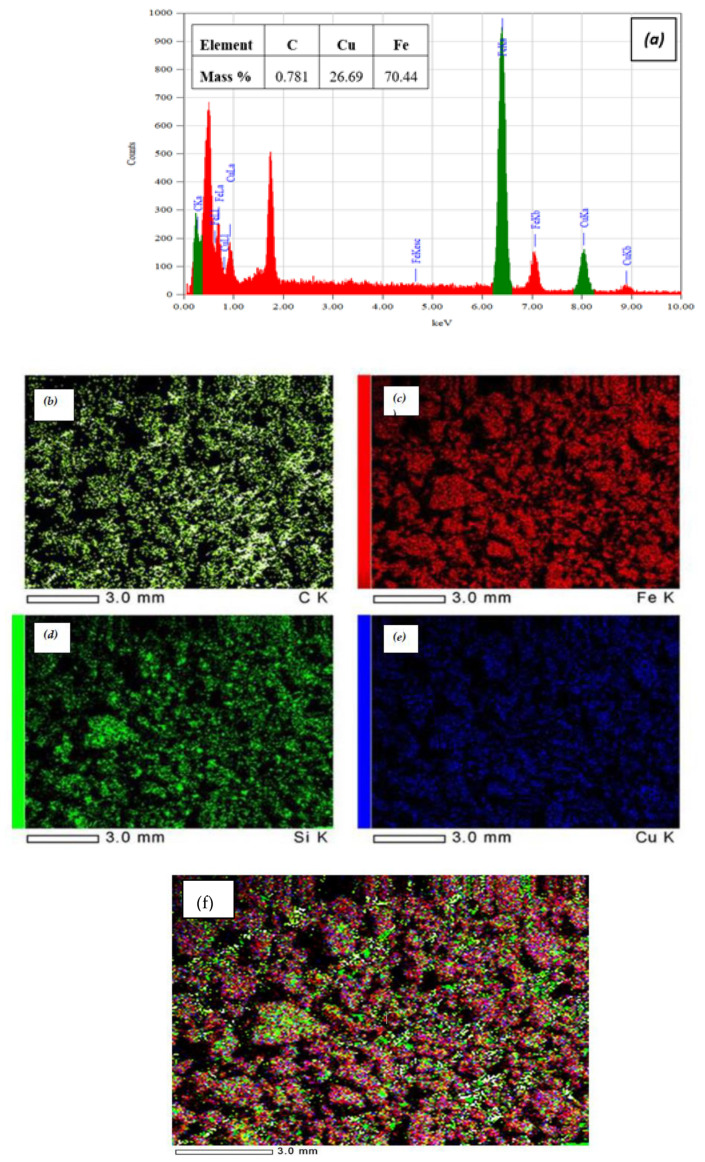
EDX analysis of LCF sample: (**a**) elemental analysis; (**b**) C; (**c**) Fe; (**d**) Si; (**e**) Cu; (**f**) mapping of the LCF.

**Figure 5 molecules-27-01831-f005:**
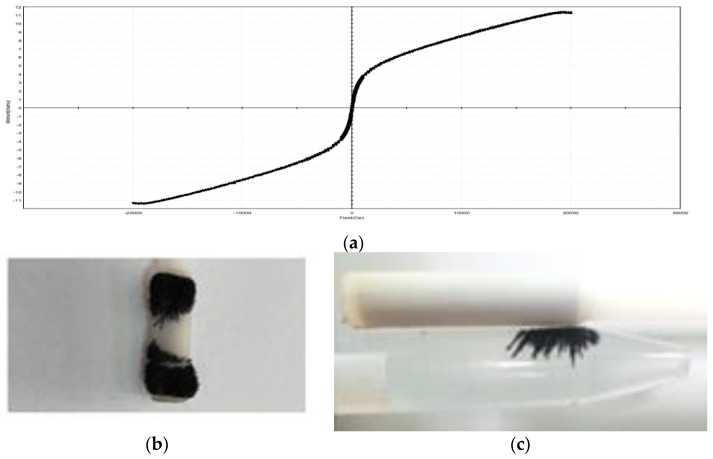
Photographs of (**a**) magnetization curve of the magnetic LCF sample obtained by VSM, (**b**) LCF powder loaded on magnetic bar, and (**c**) LCF powder in water collected with eternal magnetic bar.

**Figure 6 molecules-27-01831-f006:**
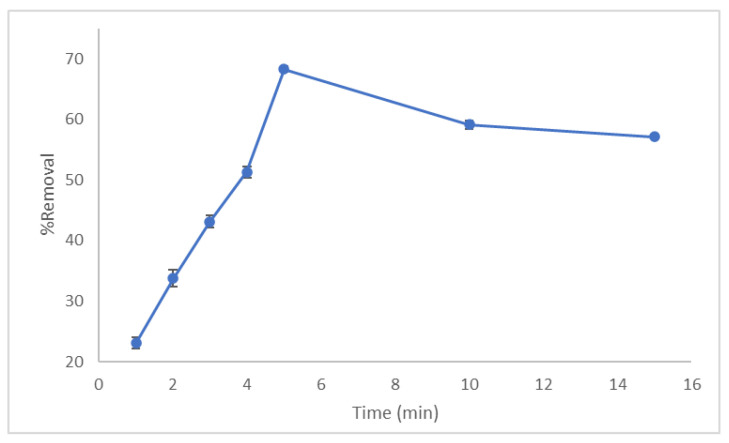
Effect of contact time on % removal of CV using LCF.

**Figure 7 molecules-27-01831-f007:**
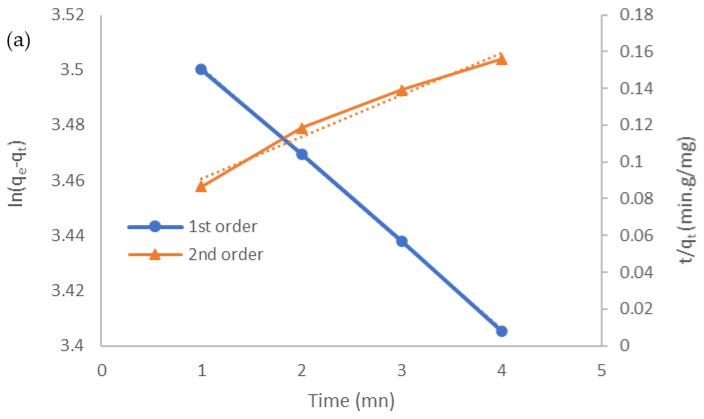
Kinetic plots for the adsorption of CV by LCF: (**a**) pseudo-first-order and pseudo-second-order models; (**b**) intraparticle diffusion model; (**c**) Elovich model.

**Figure 8 molecules-27-01831-f008:**
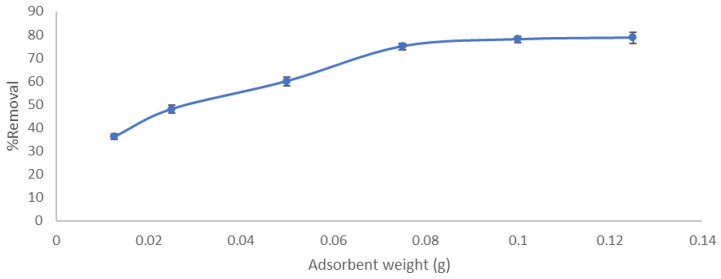
Effect of adsorbent weight on % removal of CV using LCF.

**Figure 9 molecules-27-01831-f009:**
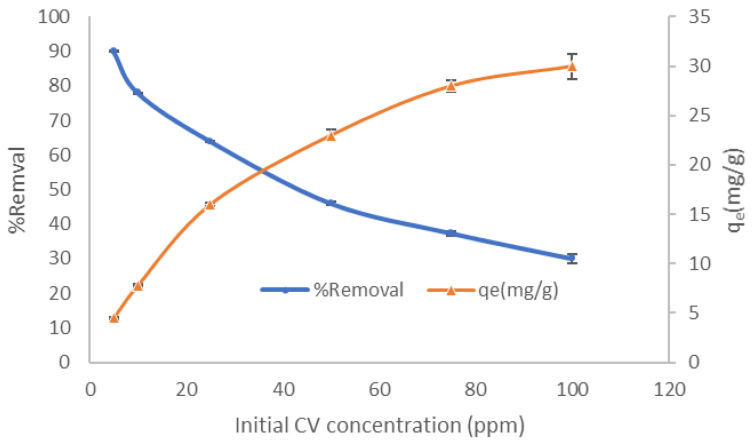
Effect of initial CV concentration on % removal and adsorption capacity using LCF.

**Figure 10 molecules-27-01831-f010:**
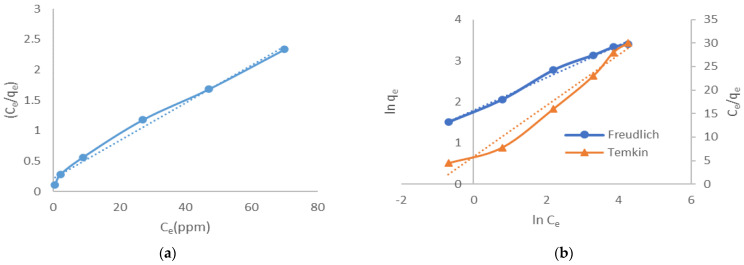
Isotherm plots for the adsorption of CV using LCF (C_0_ = 25, 50, 75, and 100 ppm; m = 0.025 g; contact time = 5 min): (**a**) Langmuir and (**b**) Freundlich and Temkin isotherm models.

**Figure 11 molecules-27-01831-f011:**
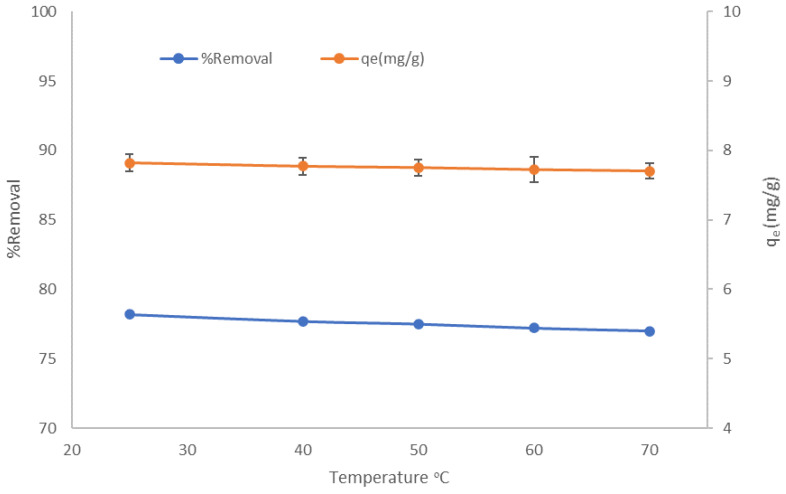
Effect of temperature on % removal and adsorption capacity using LCF.

**Figure 12 molecules-27-01831-f012:**
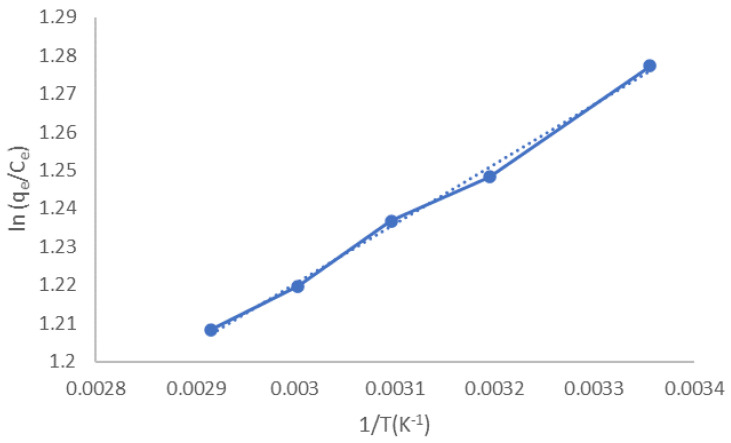
Thermodynamic plots for adsorption of CV using LCF.

**Figure 13 molecules-27-01831-f013:**
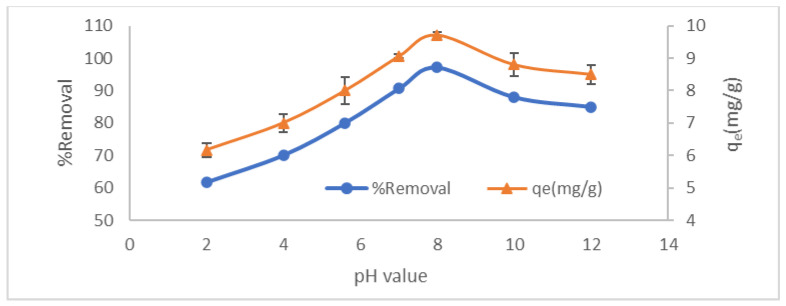
Effect of pH on % removal and LCF adsorption capacity.

**Figure 14 molecules-27-01831-f014:**
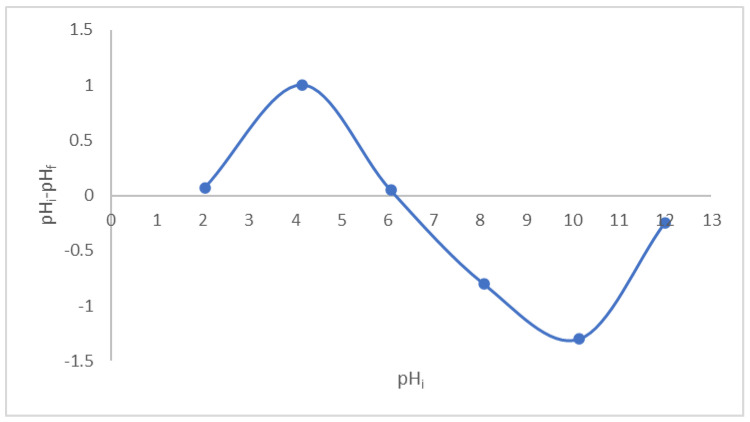
Point of zero charges of LCF adsorbent.

**Figure 15 molecules-27-01831-f015:**
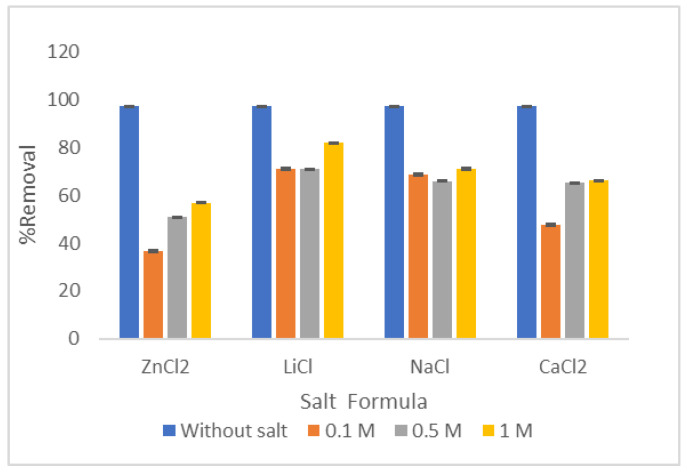
Effect of presence of different salts on % removal of CV using LCF.

**Figure 16 molecules-27-01831-f016:**
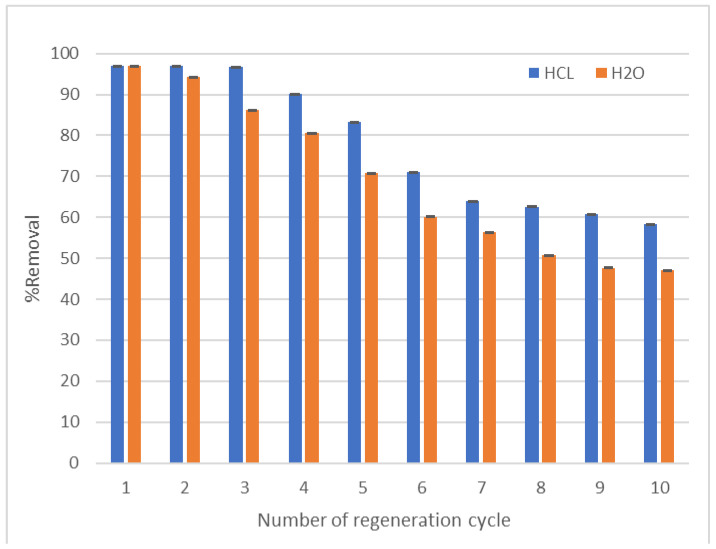
Reusability of LCF adsorbent (V = 25 mL, adsorbent weight = 0.05 g, pH = 7, agitation speed = 200 rpm, solution temperature = 25 °C, dye concentration = 10 ppm, adsorption time = 6 min and desorption time = 30 min).

**Table 1 molecules-27-01831-t001:** The experimental conditions for adsorption of CV using LCF.

Experiment Type	Concentration of CV (ppm)	Adsorption Time (min)	Adsorbent Weight (g)	pH	Temperature (°C)
Influence of time	50	5–15	0.05	7	25
Influence of adsorbent weight	10	5	0.025–0.125	7	25
Influence of CV concentration	5–100	5	0.025	7	25
Influence of pH	10	5	0.025	2–12	25
Influence of temperature	10	5	0.025	8	25–70

**Table 2 molecules-27-01831-t002:** Model parameters for adsorption kinetic study of adsorption CV using LCF.

First-Order		Second-Order		Intraparticle Diffusion		Elovich	
q_e_ (experimental)mg/g				34.129			
q_e_(mg/g)	34.2	q_e_ mg/g	14.727	C (mg/g)	6.997	β(mg/g)	21.505
K_1_(L/min)	0.0316	K_2_(g/mg·min)	0.2022	k_dif_(mg/min^0.5^·g)	17.272	α(g/g·min)	0.046
R^2^	0.9998	R_2_	0.9782	R^2^	0.954	R^2^	0.7874

**Table 3 molecules-27-01831-t003:** Model parameters for adsorption isotherm study of adsorption CV by LCF.

Langmuir		Freundlich		Temkin	
q_m_(mg/g)	30.5	n_f_	2.516	b	457.48
K_L_(L/mg)	0.1411	K_f_	6.01	K_t_	2.95
R^2^	0.9904	R^2^	0.9925	R^2^	0.09684
R_L_	0.013				

**Table 4 molecules-27-01831-t004:** Model parameters for adsorption thermodynamic study of CV adsorption using LCF.

ΔS_0_ (J/K.mol)	ΔH_0_ (KJ/mol)	ΔG_0_(Kj/mol)				
		25 °C	40 °C	50 °C	60 °C	70 °C
4.74	−1.3	−2.736	−2.7836	−2.831	−2.8784	−2.9258

**Table 5 molecules-27-01831-t005:** Comparison between LCF capacity and other adsorbents used for CV removal.

Adsorbent	Conditions	Adsorbent Capacity	Time	References
Lignin copper ferrite (LCF)	pH 7, 0.05 g adsorbent dose, 27 °C for dye initial concentration of 50 mg/L	34.12 mg/g	5 min	Present study
Natural Iraqi porcelanite rock powder	pH 8, 0.02 g adsorbent dose, 25 C for dye initial concentration of 30 mg/L	31.38 mg/g	20 min	[37]
Activated charcoal	200 ultrasonic intensity, 5 g adsorbent dose	24 mg/g	90 min.	[38]
Semiconductor nanoparticles	1.5–3 g of Nanocatalyst adsorbents in the basic medium	12.66 mg/g	100–120 min	[39]
P-type zeolite/carbon composite	initial dye concentration 100 mg/L, pH 2	11.2 mg/g	120 min	[40]
Semiconductor nanoparticles TiO_2_ with natural adsorbents	Adsorbent wight 0.5–1.2 g of nanocatalyst	9.875 mg/g	120 min	[41]
Copolymer adsorbent	pH 10, adsorbent weight 0.1 g, initial concentration 10 ppm	9.8 mg/g	180 min.	[42]
Polyacrylonitrile based-beads	pH 7, Co = 10 mg/L, adsorbent dose = 0.4 gm, 200 rpm, T = 35 °C	5.46 mg/g	300 min	[43]
Mesoporous ZnO @ silica fume-derived SiO_2_ nanocomposite	10 ppm initial concentration, at pH 9	4.9 mg/g	60 min	[44]
Date palm fiber	Adsorbent dose = 0.25 g, pH 7 at room temperature	0.66 × 10^−6^ mol g^−1^	150 min	[45]

## Data Availability

The data presented in this study are available on request from the corresponding author.

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
