# Peer review of "Lignin from Agro-Industrial Waste to an Efficient Magnetic Adsorbent for Hazardous Crystal Violet Removal"

_molecules, 2022, doi:10.3390/molecules27061831_

Round 1

Reviewer 1 Report

I’ve just finished a review of the paper molecules-1592090 titled “Lignin from agro-industrial waste to an efficient magnetic adsorbent for hazardous crystal violet dye removal” and written by authors Sara Elsayed ABD Elhafez, Zahwa Elsagan, Rehab M Ali.

In the paper removal of the crystal violet dye by magnetic adsorbent obtained from Lignin from agro-industrial waste has been investigated.

In general, the paper is interesting and easy for reading. The authors made enough experiments, in order to explain dye removal by applied material. The used material is unique and prepared in an original way, and from that point in the paper, I recognize some degree of originality.

The topic of the paper is in agreement with the topic and scope of the journal Molecules. I suggest acceptance of this paper for publication.

However, in the paper are some shortcomings and because of that, I recommend accepting the paper after major revision.

My other comments are:

  1. Lines 149-150: formulae 1 and 2 are not necessary. I suggest their removal from the paper.
  2. Line 154: add a reference for the determination of the point of zero charge
  3. Figure 2a: The bands in the range 4000-1500 cm-1 are not clear. They are very small in intensities and because of that, it is very difficult to use them to explain something or to prove the presence of a component. For that reason, the discussion for these spectral bands is not adequate and the discussion needs to be corrected.
  4. Line 237: The explanation of the spectral band at 1080 in Figure 2a is not correct. Authors should check and give explanations again by using adequate references.
  5. In Figure 2a, also add insertion of the clear LIGNITE and compare given results with it.
  6. Lines 241-244: Please add references
  7. If the FTIR spectra of the FCF and FCLA are compared, why the spectral band of water is visible in the spectrum of FLCA and not visible in FCA?
  8. Figures 7 and 8: Why on Figure 7 are 7 experimental dots, while in Figure 8 only four? In order to get a better picture of the system, correct Figure 8 and include all experimental dots, not only selected and some of them.
  9. In kinetic, intraparticle diffusion model results are missing.
  10. Also, the kinetic is determined for very low initial CV concentration which is significantly lower in comparison with maximal adsorption capacity (Figure 9). For that reason, obtained results are not good and useful. Authors must check and adapt discussion or insert new results which much more describe the system.
  11. Figures 10 and 11 are not obtained from the same number of experimental dots. Because of that Figure 11 does not explain the system well. It is required from authors to correct it.
  12. From thermodynamic authors conclude that physisorption is present in the system. From kinetic, authors mentioned chemisorption, while isotherm study showed that Langmuir model may be used as the best model for describing the system, which is not in agreement. That is perhaps a consequence of obtaining Figures with a non-adequate number of experimental dots. It is required from authors to adopt discussion after new Figures are added. Also, the explanation for changes in entropy is not good and disagrees with other results.
  13. From pH measuring authors also include electrostatic interactions, what additional complicate system and additionally is not in agreement with Langmuir model, physisorption, chemisorption. The discussion must be changed. Also, an explanation for the best adsorption at pH8 is missing.
  14. In Figure 17 it is not clear what represents the Series. Please be more specific.
  15. In Table 4, please add all results in the same units.

Best regards

Author Response

My other comments are:

  1. Lines 149-150: formulae 1 and 2 are not necessary. I suggest their removal from the paper.

The two equation were deleted

  1. Line 154: add a reference for the determination of the point of zero charge

The reference was added

  1. Figure 2a: The bands in the range 4000-1500 cm-1 are not clear. They are very small in intensities and because of that, it is very difficult to use them to explain something or to prove the presence of a component. For that reason, the discussion for these spectral bands is not adequate and the discussion needs to be corrected.

The band in this region is represent the present of the lignin which was mixed in very small amount and this explanation was clarify in discussion

  1. Line 237: The explanation of the spectral band at 1080 in Figure 2a is not correct. Authors should check and give explanations again by using adequate references.

The explanation was corrected

  1. In Figure 2a, also add insertion of the clear LIGNITE and compare given results with it.

The present of lignin was provided by the small bands at 2047.28 cm-1 and 1543.36 cm-1

  1. Lines 241-244: Please add references

The reference was added

  1. If the FTIR spectra of the FCF and FCLA are compared, why the spectral band of water is visible in the spectrum of FLCA and not visible in FCA?

This peak maybe detected in the LCFA sample due to insufficient drying after the adsorption process

  1. Figures 7 and 8: Why on Figure 7 are 7 experimental dots, while in Figure 8 only four? In order to get a better picture of the system, correct Figure 8 and include all experimental dots, not only selected and some of them.

I was able to apply all points in the Figure 7 in Figure only on the 2nd order model, However the rest of the studied models, it is not possible to apply the point that qt =qe and therefore only 5 pints were applied. The Figures were modified

  1. In kinetic, intraparticle diffusion model results are missing.

It presented in Figure 8(b) and in Table 1

  1. Also, the kinetic is determined for very low initial CV concentration which is significantly lower in comparison with maximal adsorption capacity (Figure 9). For that reason, obtained results are not good and useful. Authors must check and adapt discussion or insert new results which much more describe the system.

section 3.2.1 was repeated with higher concentration

  1. Figures 10 and 11 are not obtained from the same number of experimental dots. Because of that Figure 11 does not explain the system well. It is required from authors to correct it.

Figure 11(a) and (b) were corrected

  1. From thermodynamic authors conclude that physisorption is present in the system. From kinetic, authors mentioned chemisorption, while isotherm study showed that Langmuir model may be used as the best model for describing the system, which is not in agreement. That is perhaps a consequence of obtaining Figures with a non-adequate number of experimental dots. It is required from authors to adopt discussion after new Figures are added. Also, the explanation for changes in entropy is not good and disagrees with other results.

This inconsistency in the results was corrected after restating the figures and taking into account more points to express the system more accurately

  1. From pH measuring authors also include electrostatic interactions, what additional complicate system and additionally is not in agreement with Langmuir model, physisorption, chemisorption. The discussion must be changed. Also, an explanation for the best adsorption at pH8 is missing.

It has been proven that the system follows Langmuir and Freundlich as well, and this was proven after re-analysis using a more accurate method. The explanation of the reason that pH=8 is the best, described in the pHpzc paragraph

  1. In Figure 17 it is not clear what represents the Series. Please be more specific.

corrected

  1. In Table 4, please add all results in the same units.

corrected

Reviewer 2 Report

In review, I received a manuscript entitled "Lignin from agro-industrial waste to an efficient magnetic ad-2 sorbent for hazardous crystal violet removal"  considered for publication in MDPI journal  Molecules. 

The manuscript presents the results of Crystal Violet (CV) removal from aqueous solutions by utilising a magnetic novel adsorbent produced from wasted lignin. Below, I summarize a few comments and suggestions for further improvement of the manuscript before publication:

Abstract should be improved by adding some quantitative data, like adsorption capacity.

Please check the typos in the whole manuscript … overall, the grammar throughout the article should be carefully revised. There are some sentences that possibly needs rewording, word spacing needs to be checked, as does punctuation. For instance, in line 18 there is a "of5," it need to be separated. In line 31, there is a dot before Introduction.

Please, I urge you to carefully review the journal's editorial guidelines. References do not meet the suggested standard.  For example, in line 43 references should be presented as [4-6].

Please, unify the presentation of chemicals, they are stated by name and formula in brackets, but there are some whose formula is presented without being in parentheses.

Please unify, accordingly the the journal's editorial guidelines there should  be a space between number and unit.

In section "2.4 Adsorption investigates" you need to put a table with a experimental design. Also, the subtitle is veri confusing. 

Lines 141-142: "After different time intervals, specific quantities were suction from the reaction media by 141 filter syringe" How often did you exclude the sample? Which volume? How do you guarantee constant volume?

Please, review the jorunal's editorial guidelines for the presentation of equations. 

Line 156: "pHpzc" that should be a subscript.

Regarding "2. Materials and methodology," this section should be referenced unless, the methods presented have been developed by the authors.

Line 223: Entropy units are not KJ/mol. Please, revise. 

In FTIR, the changes induced by CV binding as evidenced by the IR spectra would be easier to follow if both curves are presented in one plot and normalised. It would be also better if the differential spectrum be given. In the current version of Figure 2 one cannot see any substantial changes.

It would be better if the figures were presented close to the text where the results are discussed.

Table 3 reports thermodynamic parameters of the process. Data in the first line does not fulfil the thermodynamic formula: ΔG = ΔH -TΔS. The initial conditions would be put in Table or in Table caption. 

Table 4 shows a summary of different adsorbents used in CV removal in previous studies, data should be presented in terms of adsorption capacity with the same units. And I suggest that they be organized from largest to smallest. As presented, they are not comparable.

I suggest performing a statistical analysis of the data. Likewise, the presentation of the standard deviation in figures 7, 9, 12, 14, 16, 17, and 18.

Results must be compared with previous studies. 

Author Response

Abstract should be improved by adding some quantitative data, like adsorption capacity.

Done

Please check the typos in the whole manuscript … overall, the grammar throughout the article should be carefully revised. There are some sentences that possibly needs rewording, word spacing needs to be checked, as does punctuation. For instance, in line 18 there is a "of5," it need to be separated. In line 31, there is a dot before Introduction.

The manuscript was checked

Please, I urge you to carefully review the journal's editorial guidelines. References do not meet the suggested standard.  For example, in line 43 references should be presented as [4-6].

corrected

Please, unify the presentation of chemicals, they are stated by name and formula in brackets, but there are some whose formula is presented without being in parentheses.

corrected

Please unify, accordingly the the journal's editorial guidelines there should  be a space between number and unit.

corrected

In section "2.4 Adsorption investigates" you need to put a table with a experimental design. Also, the subtitle is veri confusing. 

Done

Lines 141-142: "After different time intervals, specific quantities were suction from the reaction media by 141 filter syringe" How often did you exclude the sample? Which volume? How do you guarantee constant volume?

I prepared for each tested time one flask and take three samples from the same flask to  verify the validity of the results. This point was clarifying in the manuscript

Please, review the jorunal's editorial guidelines for the presentation of equations. 

corrected

Line 156: "pHpzc" that should be a subscript.

corrected

Regarding "2. Materials and methodology," this section should be referenced unless, the methods presented have been developed by the authors.

The reference was added

Line 223: Entropy units are not KJ/mol. Please, revise. 

corrected

In FTIR, the changes induced by CV binding as evidenced by the IR spectra would be easier to follow if both curves are presented in one plot and normalised. It would be also better if the differential spectrum be given. In the current version of Figure 2 one cannot see any substantial changes.

The two figure was modified to be in one figure

It would be better if the figures were presented close to the text where the results are discussed.

done

Table 3 reports thermodynamic parameters of the process. Data in the first line does not fulfil the thermodynamic formula: ΔG = ΔH -TΔS. The initial conditions would be put in Table or in Table caption. 

corrected

Table 4 shows a summary of different adsorbents used in CV removal in previous studies, data should be presented in terms of adsorption capacity with the same units. And I suggest that they be organized from largest to smallest. As presented, they are not comparable.

This table was modified

I suggest performing a statistical analysis of the data. Likewise, the presentation of the standard deviation in figures 7, 9, 12, 14, 16, 17, and 18.

Done

Results must be compared with previous studies. 

Done

Round 2

Reviewer 1 Report

I've finished a review of the second version of the manuscript. 

In the new version of the manuscript, the authors did not clearly mark all changes which were made. Please clearly point out all the changes that have been made in the paper, not just some of them.

Also, on questions or suggestions, authors must give more complete answers and make deeper changes in the paper. It is not enough if only cosmetic changes were made, and added one or two superficial sentences, especially for the part of the text where it was suggested that the text was wrong or without proper explanations. 

In the new version of the paper, some references were not cited adequalty. some Figure captions are not adequate for new figures.

Please make a detailed correction and check the paper again, and then send the final version. 

Author Response

Please find the attaced file for my response for the reviewer comments
